# Precise radiometric age establishes Yarrabubba, Western Australia, as Earth's oldest recognised meteorite impact structure

Timmons M. Erickson [1,2,3]*, Christopher L. Kirkland [4], Nicholas E. Timms[2], Aaron J. Cavosie[2] & Thomas M. Davison [5]

The ~70 km-diameter Yarrabubba impact structure in Western Australia is regarded as among Earth's oldest, but has hitherto lacked precise age constraints. Here we present U–Pb ages for impact-driven shock-recrystallised accessory minerals. Shock-recrystallised monazite yields a precise impact age of 2229 ± 5 Ma, coeval with shock-reset zircon. This result establishes Yarrabubba as the oldest recognised meteorite impact structure on Earth, extending the terrestrial cratering record back >200 million years. The age of Yarrabubba coincides, within uncertainty, with temporal constraint for the youngest Palaeoproterozoic glacial deposits, the Rietfontein diamictite in South Africa. Numerical impact simulations indicate that a 70 km-diameter crater into a continental glacier could release between $8.7 \times 10^{13}$ to $5.0 \times 10^{15}$ kg of $H_2O$ vapour instantaneously into the atmosphere. These results provide new estimates of impact-produced $H_2O$ vapour abundances for models investigating termination of the Paleoproterozoic glaciations, and highlight the possible role of impact cratering in modifying Earth's climate.

[1] Jacobs—JETS, Astromaterials Research and Exploration Science Division, NASA Johnson Space Center, 2101 NASA Parkway, Houston, TX 77058, USA. [2] The Institute for Geoscience Research (TIGeR), Space Science and Technology Centre, School of Earth and Planetary Sciences, Curtin University, GPO Box 1984, Perth, WA 6845, Australia. [3] Center for Lunar Science and Exploration, Lunar and Planetary Institute, Universities Space Research Association, 3600 Bay Area Blvd, Houston, TX 77058, USA. [4] The Institute for Geoscience Research (TIGeR), Centre for Exploration Targeting—Curtin Node, School of Earth and Planetary Sciences, Curtin University, GPO Box 1984, Perth, WA 6845, Australia. [5] Impacts and Astromaterials Research Centre, Department of Earth Science and Engineering, Imperial College London, London SW7 2AZ, UK. *email: Timmons.M.Erickson@nasa.gov

Extraterrestrial bombardment flux is speculated to have had major consequences for the development of Earth's surface environment[1,2]. However, the terrestrial impact record is fragmentary, principally due to tectonics and erosion[3,4], and is progressively erased into the geologic past when, conversely, the bombardment rate was larger than today[5]. The oldest record of impacts on Earth are Archaean to Palaeoproterozoic ejecta deposits found within the Kaapvaal craton of southern Africa and the Pilbara Craton in Western Australia, spanning ca. 3470 (ref. [6]) to 2460 Ma[7]; however, no corresponding impact craters have been identified. Currently only two precisely dated Precambrian-age impact structures are known, the 2023 ± 4 Ma, >250 km Vredefort Dome in South Africa[8,9], and the 1850 ± 1 Ma, >200 km Sudbury structure in Canada[10]. Other purported Palaeoproterozoic-age impact structures have either poorly constrained ages[11] or highly contentious impact evidence[12,13].

A consequence of the incomplete terrestrial impact record is that connections between impact events and punctuated changes to the atmosphere, oceans, lithosphere, and life remain difficult to establish, with the notable exception of the Cretaceous–Paleogene impact[14,15]. Hitherto, the impact cratering record was absent from 2.5–2.1 Ga, when significant changes in the Earth's hydrosphere and atmosphere occurred[16,17].

Yarrabubba is a recognised impact structure located within the Murchison Domain of the Archaean granite—greenstone Yilgarn Craton of Western Australia (Fig. 1)[18]. No circular crater remains at Yarrabubba; however, the structure has an elliptical aeromagnetic anomaly consisting of an even, low total magnetic intensity domain, measuring approximately 20 km N–S by 11 km E–W (Fig. 1)[18]. The present day exposure represents a deep erosional level, as neither impact breccias nor topographic expressions of the over-turned rim or central uplift are preserved. Therefore, the ~20 km diameter magnetic anomaly has been interpreted to represent the remnant of the deeply buried central uplift of the structure, which is consistent with an original crater diameter of 70 km[18,19]. Unshocked dolerite dykes formed during either ca. 1200 Ma Muggamurra[20] or ca. 1075 Ma Warakurna[21] regional volcanism cross-cut the elliptical magnetic anomaly and thus post-date the impact event.

The main target rocks at the Yarrabubba structure are granitoids collectively known as the Yarrabubba monzogranite (Fig. 1). Identification of shocked quartz and shatter cones in the Yarrabubba monzogranite confirmed an impact origin for the structure[18,22]. The structure is centred on a large exposure of granophyre known locally as Barlangi Rock (Fig. 1; 118˚50′E, 27˚10′S). Barlangi granophyre is a sodic rhyolite[22] that has been interpreted as an impact-generated melt rock[18], radiating dyke-like apophyses of granophyre outcrop as far as 3 km from the centre of the structure. The Barlangi granophyre has thus been interpreted to have intruded into the Yarrabubba monzogranite along faults rather than forming a flat-lying, crater-filling melt sheet, similar to metanorite dykes and apophyses interpreted as impact melt that are exposed in the core of the deeply eroded Vredefort impact structure[23].

The age of the Yarrabubba impact structure was previously constrained only to be younger than the 2650 ± 10 Ma Yarrabubba monzogranite[24] and older than the ca. 1200-1075 Ma cross-cutting dolerite dykes. Zircon crystals from the Barlangi granophyre have previously yielded a complex age spectrum that span nearly 500 Myr, from 2.79 to 2.23 Ga[24,25]. Pseudotachylite veins at Yarrabubba yield a sericite $^{39}$Ar/$^{40}$Ar age of ca. 1.13 Ga[26], which likely records alteration during younger mafic volcanism.

This study utilises targeted in situ U–Pb geochronology by secondary ion mass spectrometry (SIMS) to analyse recrystallised domains (neoblasts) in monazite and zircon, which have been shown to yield precise ages for ancient impact events[27–29]. We

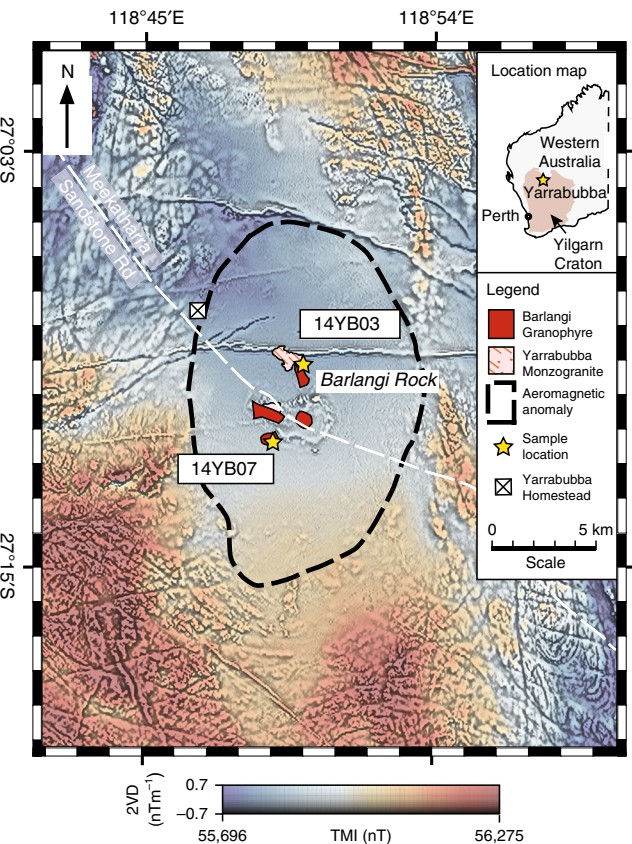

**Fig. 1 Map of the Yarrabubba impact structure and sample localities.** Composite aeromagnetic anomaly map of the Yarrabubba impact structure within the Yilgarn Craton, Western Australia, showing the locations of key outcrops and samples used in this study. The image combines the total magnetic intensity (TMI, cool to warm colours) with the second vertical derivative of the total magnetic intensity (2VD, greyscale) data[81]. The demagnetised anomaly centred on the outcrops of the Barlangi granophyre is considered to be the eroded remnant of the central uplift domain, which forms the basis of the crater diameter of 70 km[18]. Prominent, narrow linear anomalies that cross-cut the demagnetised zone with broadly east-west orientations are mafic dykes that post-date the impact structure.

present high-resolution orientation mapping and correlated in situ U–Pb analysis to investigate the microstructure and age of shock features in zircon and monazite in target rock and impact melt from the Yarrabubba structure in Western Australia. These results establish Yarrabubba as the oldest preserved impact structure on Earth.

## Results and discussion

**Zircon and monazite shock microstructures.** Within the Yarrabubba monzogranite, zircon and monazite grains preserve a range of impact-related microstructures. Zircon displays primary igneous growth zoning that is cross-cut by planar and subplanar shock microstructures, including {112} shock twins and {100} planar deformation bands (Fig. 2a; Supplementary Fig. 1)[30–32]. Monazite preserves a broader range of impact-related textures including domains with low-angle subgrain boundaries and multiple sets of shock twins along (001), (100) and (101), and domains of strain-free neoblasts (Fig. 2b; Supplementary Fig. 2)[28,33].

In Barlangi granophyre, zircon textures range from unshocked grains preserving primary igneous growth zoning, to grains containing clear evidence of impact metamorphism, such as planar microstructures, polycrystalline aggregates and grains with $ZrO_2$ inclusions (Fig. 2c; Supplementary Fig. 3)[34]. Neoblasts

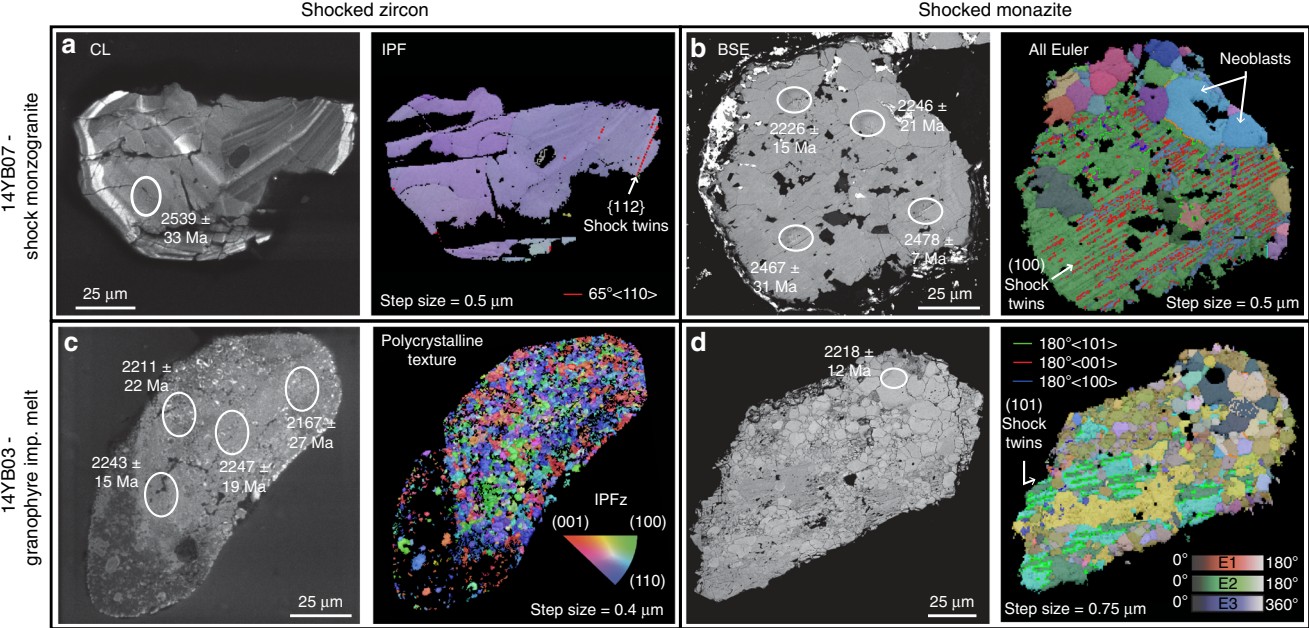

**Fig. 2 Monazite and zircon shock recrystallisation textures.** Examples of shocked zircon and monazite grains from Yarrabubba monzogranite sample 14YB07 and Barlangi granophyre impact melt sample 14YB03. **a** Cathodoluminescence (CL) and Inverse Pole Figure (IPF) images of a shocked zircon with {112} deformation twins. The zircon contains primary oscillatory zoning that is cross-cut by shock deformation twins and subplanar low-angle grain boundaries. **b** Backscattered electron (BSE) image and electron backscatter diffraction (EBSD) all Euler map of a shocked monazite with systematic shock twin domains that are overprinted by neoblasts. **c** CL and IPF images of a polycrystalline shocked zircon grain from the granophyric impact melt. Note that individual crystallites exhibit concentric CL zonation patterns while the overall CL pattern reflects the original zonation of the grain with a CL brighter core to CL dark rim. The IPF orientation map of the zircon is dominantly blue to pink and many of the granules have systematic grain boundaries of either 65°/ <110> or 90°/<110>. **d** BSE image and all Euler map of shock-deformed monazite displaying highly deformed and twinned domains that are overprinted by neoblasts, from the Barlangi granophyre. Location of U–Pb SIMS analytical spots are denoted on each grain with the $^{207}Pb/^{206}Pb$ age and $2\sigma$ errors in Ma.

within polycrystalline zircon aggregates contain systematic misorientation relationships with one another of either 65° about <110> or 90° about <110>, which can only be caused by recrystallisation after formation of {112} twins and the high-pressure polymorph reidite, respectively, and are unambiguous indicators of shock metamorphism[34–36]. While the $ZrO_2$ inclusions index as baddeleyite (monoclinic-$ZrO_2$), crystallographic orientation relationships among transformation twins demonstrate they originally formed from tetragonal-$ZrO_2$ parent grains[37]. Thermal dissociation of zircon to tetragonal-$ZrO_2$ only occurs in silica-saturated melts above 1673 °C[34], unequivocally indicating the Barlangi granophyre was a super-heated impact melt. Monazite grains in Barlangi granophyre preserve a similar range of impact-related features to those from Yarrabubba monzogranite, including crystal-plastic strain, deformation twins diagnostic of shock conditions, and strain-free neoblastic domains (Fig. 2d; Supplementary Fig. 4).

**Shocked zircon and monazite U–Pb results.** The U–Pb SIMS analyses of zircon from the Yarrabubba monzogranite define a regression, which intercepts concordia at 2626 ± 36 and 1202 ± 210 Ma (mean square weight of deviates [MSWD] = 1.8; Supplementary Table 1). The upper intercept is interpreted as the primary magmatic crystallisation age of the target rocks, which is within uncertainty of the 2650 ± 20 Ma (n = 6) age obtained previously[24]. The lower intercept age is attributed to partial resetting associated with post-impact dolerite intrusion in the Mesoproterozoic[20,21]. Monazite $^{207}Pb/^{206}Pb$ ages from Yarrabubba monzogranite yield a bimodal distribution. Analytical spots from the high-strain shocked host and/or twin domains are variably discordant and record $^{207}Pb/^{206}Pb$ ages from 2478 ± 14

to 2323 ± 16 Ma (Fig. 3a, Supplementary Table 2). These ages may represent either formation during a post-crystallisation metamorphic event or partial radiogenic Pb-loss during the impact event or a subsequent thermal event. In contrast, spots from low-strain, randomly oriented neoblasts cluster around concordia, and have a weighted mean $^{207}Pb/^{206}Pb$ age of 2227 ± 5 Ma (n = 12, MSWD = 0.89) (Fig. 3b).

Barlangi granophyre zircon $^{207}Pb/^{206}Pb$ ages also show a bimodal age distribution. We interpret the oscillatory-zoned cores with apparent ages of 2781 ± 14 to 2319 ± 28 Ma to represent inherited (pre-impact) zircon grains that were incorporated into the Barlangi granophyre as xenocrysts, consistent with zircon ages determined previously[24,25]. These results indicate the presence of a significant source component in the Barlangi granophyre that predates the 2.65 Ga Yarrabubba monzogranite. Individual analyses from polycrystalline zircon domains are variably discordant and yield $^{207}Pb/^{206}Pb$ ages from 2259 ± 30 to 2156 ± 52 Ma (Supplementary Table 1). We interpret the data array (Fig. 3a) to be a function of near-recent Pb-loss resulting from exposure to surface fluids. The data array from recrystallised zircon domains yields an upper intercept age of 2246 ± 17 Ma (n = 13, MSWD = 1.2), and two rim analyses with baddeleyite intergrowths are collinear with this regression. We interpret the upper intercept to reflect both new zircon growth and near complete resetting of U–Pb systematics in pre-existing domains during shock metamorphism. This date is within uncertainty of a single previously reported U–Pb zircon analysis from the Barlangi granophyre of 2234 ± 28 Ma[24], which was inferred to indicate a Palaeoproterozoic impact age[18].

Barlangi granophyre monazite $^{207}Pb/^{206}Pb$ ages preserve a bimodal distribution similar to monazite from Yarrabubba monzogranite. Analyses from the highly strained host and

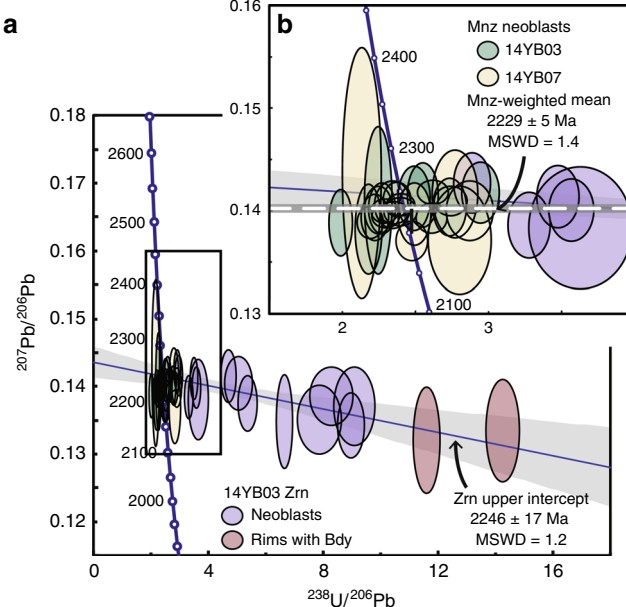

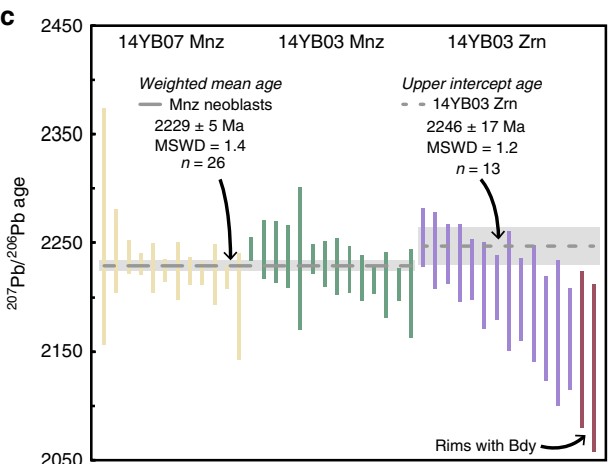

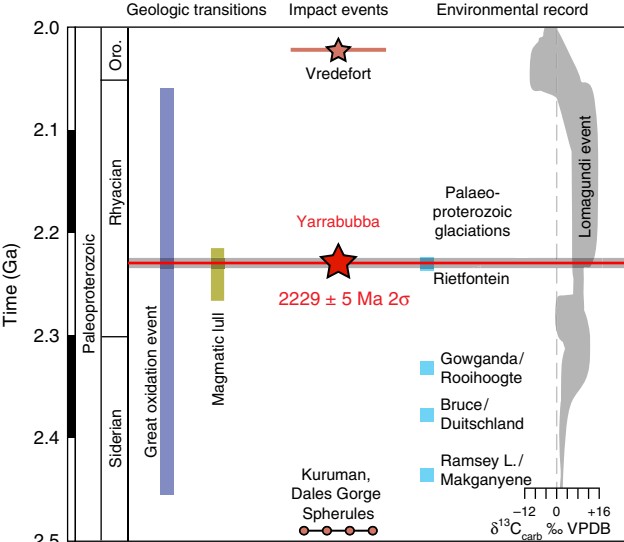

**Fig. 4 Temporal evolution of the early Palaeoproterozoic Earth.** Key features include the Yarrabubba and other impact events, the Great Oxidation Event (2.06–2.45 Ga) and glaciations (2.23–2.54 Ga). Glacial deposition age constraints from ref. [38], the global $\delta^{13}$C trend of carbonates is modified from ref. [51], and the lull in magmatic activity is modified from ref. [44]. Note the close association of the Yarrabubba impact event to the end of the final Palaeoproterozoic glaciation, the Rietfontein, at 2225 ± 3 Ma and followed by the large positive $\delta^{13}$C excursion known as the Lomagundi event[82,83]. Other impacts include the 2.02 Ga Vredefort Dome[8], and the 2.49 Ga correlated Kuruman spherule layer in the Griqualand West basin of South Africa and the Dales Gorge spherule layer of the Hamersley basin in Western Australia[7].

When combined, all neoblastic monazite domains from both Barlangi granophyre and Yarrabubba monzogranite define a cluster centred on concordia (Fig. 3a) and yield a weighted mean $^{207}$Pb/$^{206}$Pb age of 2229 ± 5 Ma ($n = 26$, MSWD = 1.4, Fig. 3c), which we interpret to record monazite recrystallisation during shock metamorphism and the best estimate of the Yarrabubba impact event (Fig. 3). The weighted mean $^{207}$Pb/$^{206}$Pb age for neoblastic zircon of 2246 ± 17 Ma ($n = 13$, MSWD = 1.2) overlaps with the monazite age, but is less precise. The new Yarrabubba impact age of 2229 ± 5 Ma determined here extends the terrestrial record of impact craters by 200 million years[8], and demonstrates the potential for discovery of ancient impact structures on Archaean cratons.

**Implications for the Palaeoproterozoic Earth**. The age constraints presented here establish Yarrabubba as the first recognised meteorite impact to have occurred during the Rhyacian period, a dynamic time in the evolution of Earth following the transition from the Archaean to the Proterozoic eon. At least four glacial diamictite deposits, three of which are found on multiple cratons, are recognised between 2.4 and 2.2 Ga[38]. Of these deposits, the >2.42 Ga Makganyene diamictite from the Kaapvaal craton of southern Africa has been interpreted to represent low-latitude glaciers that may signify global ice conditions[39]. The youngest Palaeoproterozoic glacial deposit, the Rietfontein diamictite within the Transvaal basin of South Africa, has a minimum depositional age of 2225 ± 3 Ma based on the overlying Hekpoort basalt (Fig. 4)[38,40], which is within analytical uncertainty of the Yarrabubba impact event. Glacial diamictite deposits do not appear again in the geological record for >400 million

**Fig. 3 Shock-recrystallised zircon and monazite age spectra. a** Zircon (Zrn) and monazite (Mnz) U–Pb Tera-Wasserburg concordia plot of shock-recrystallised zircon and monazite neoblasts from the Barlangi granophyre and the Yarrabubba monzogranite. Monazite analyses from the Yarrabubba monzogranite are coloured yellow, monazite analyses from the Barlangi granophyre are coloured green, neoblastic zircon analyses from the granophyre are blue and analyses from zircon rims with baddeleyite (Bdy) intergrowths are maroon. While all zircon neoblastic domains show variable Pb-loss, the analyses form a linear discordia to ca. 2240 Ma, interpreted as the impact age. **b** All monazite neoblasts record a weighted mean $^{207}$Pb/$^{206}$Pb age of 2229 ± 5 Ma (horizontal red bar). **c** Weighted mean $^{207}$Pb/$^{206}$Pb age plot of the neoblastic domains from monazite grains of the shock-deformed Yarrabubba monzogranite (yellow), monazite from the Barlangi granophyre (green) and zircon from the granophyre (blue). Note overlap of weighted mean age from each monazite population, supporting the interpretation of 2229 ± 5 Ma as the impact age. All error ellipses and bars are reported at the 2$\sigma$ level.

twinned domains display variable normal and reverse discordance and record $^{207}$Pb/$^{206}$Pb ages between 2457 ± 24 and 2284 ± 14 Ma. In contrast, analyses from low-strain, randomly oriented neoblasts cluster around concordia, with a weighted mean $^{207}$Pb/$^{206}$Pb age of 2231 ± 8 Ma ($n = 14$, MSWD = 1.9) (Fig. 3b; Supplementary Table 2).

years (Fig. 4)[41,42]. What caused the extended absence of glacial conditions after ca. 2225 Ma is debated[43]. The end of the Palaeoproterozoic glaciations at 2225 Ma occurred within an apparent ~50 Myr lull in global magmatism from 2266 to 2214 Ma[44], making it difficult to appeal to volcanic outgassing as having played a significant role in forcing the glacial termination. Therefore, other mechanisms such as impact cratering need consideration. Radiometric age data presented here demonstrate synchronicity, within uncertainty, between the $2229 \pm 5$ Ma Yarrabubba impact event and the termination of glacial conditions (i.e. Rietfontein diamictite) at $2225 \pm 3$ Ma. The geographic extent of the Rietfontein diamictite is poorly constrained, and it is not yet known if global glacial conditions existed at this time. Nonetheless, we apply numerical simulations below to explore the potential effects that a Yarrabubba-sized impact may have had on climactic conditions.

Several factors caused by the Yarrabubba impact event could have triggered a change in regional or global climate. Depending on the ambient climate state and palaeogeographic nature of the northern Yilgarn craton at the time of impact (e.g., ice cover, shallow ocean or carbonate platform overlying silicate basement), which is unknown, significant amounts of $CO_2$, water vapour or other greenhouse gases could have been released into the relatively oxygen-poor Palaeoproterozoic atmosphere[45] by the impact event. Given that the age of the Yarrabubba impact overlaps with the youngest Paleoproterozoic glacial deposits, we explore scenarios where the Yarrabubba impact site could have been covered by a continental ice sheet at the time of impact. Numerical models using the iSALE shock physics code[46–48] (see Methods) demonstrate that the formation of a 70-km-diameter impact crater into a granitic target with an overlying ice sheet ranging from 2 to 5 km in thickness results in the almost instantaneous vaporisation of 95–240 km$^3$ of ice and up to 5400 km$^3$ total melting (Fig. 5). The vapourised ice corresponds to between $9 \times 10^{13}$ and $2 \times 10^{14}$ kg of water vapour being jetted into the upper atmosphere within moments of the impact (Fig. 5d). Impact-generated water vapour in the lower atmosphere would have condensed and rapidly precipitated as rain and snow with no significant long-term climate effects, or could have even triggered widespread glacial conditions via cloud albedo effects during interglacial periods[49]. However, ejection of high-altitude water vapour has potential for greenhouse radiative forcing, depending critically on atmospheric residence time[50]. Nonetheless, uncertainties in the structure and composition of Earth's Palaeoproterozoic upper atmosphere mean that the precise nature of atmospheric interactions of the collapsing vapour plume is inherently difficult to model[50]. Nevertheless, considering that Earth's atmosphere at the time of impact contained only a fraction of the current level of oxygen[51], a possibility remains that the climatic forcing effects of $H_2O$ vapour released instantaneously into the atmosphere through a Yarrabubba-sized impact may have been globally significant. Understanding the residence times of impact-produced water vapour in a cold Palaeoproterozoic atmosphere, e.g. ref. [50], and the complex interplay of radiative versus insulative effects of clouds, e.g. ref. [52], during glacial conditions requires further investigation. The effects of impact cratering have long been recognised as drivers of climate change[2,53]. Many studies have described the atmospheric effects of the end-Cretaceous Chicxulub impact structure in Mexico[49,54,55], which resulted in global cooling of oceans and production of widespread acidic rains[56,57]. While the Yarrabubba structure dated at $2229 \pm 5$ Ma represents the Earth's oldest dated impact crater, its coincidence with termination of Palaeoproterozoic glacial conditions prompts further consideration of the ability of meteorite impacts to trigger climate change.

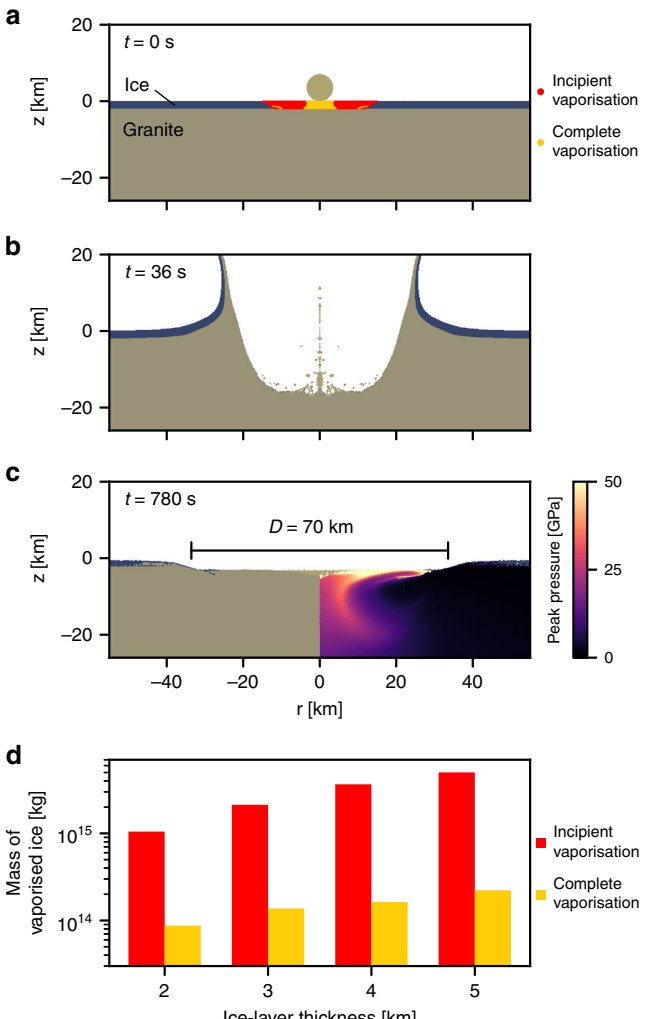

**Fig. 5 Evolution of an impact crater in an ice covered continent.** Snapshots of the iSALE model with a 2-km-thick ice sheet showing **a** the initial conditions, **b** the transient crater and **c** the final crater. Superimposed on **a** is the initial position of tracer particles which were shock-heated to the critical entropy required to begin vaporisation (incipient vaporisation, red) and to completely vapourise ice (complete vaporisation, yellow). The colour scale on the right-hand side of **c** shows the peak shock pressure in the granite basement. **d** The calculated mass of ice shock-heated to the critical entropy for incipient and complete vaporisation, as a function of initial ice thickness. In each impact, the impactor size was 7 km and resulted in a final crater diameter of ~70 km.

## Methods
**Samples and preparation**. Two samples from the Yarrabubba impact structure were selected for analysis (Fig. 1): a sample of the shocked Yarrabubba monzogranite (14YB07) and a sample from the Barlangi granophyre (14YB03). The Yarrabubba monzogranite was sampled from a small outcrop approximately 1.2 km WSW of Barlangi Rock. The Barlangi granophyre was sampled 2.7 km NNW of Barlangi Rock from an apophyse intruding the Yarrabubba monzogranite along what appears to be a shallowly dipping fault plane or fracture.

Thin sections were prepared from each hand sample to allow petrographic characterisation of the lithological fabric and identification of shock features. Zircon and monazite grains were separated from each sample. To separate zircon and monazite grains ~1 kg splits of each sample were processed with a Selfrag electric pulse disaggregator in the John de Laeter Centre (JdLC), Curtin University, Western Australia. The heavy mineral fraction was then separated using the heavy liquid methylene iodide. Further concentration of zircon and monazite was achieved with a Frantz isodynamic magnetic separator. Grains were then handpicked and mounted in a 25.4 mm epoxy round. The epoxy rounds were given a mechanical polish to 1 μm with diamond paste before a final chemical–mechanical polish with a colloidal dispersion of 5 nm silica in NaOH.

After polishing, monazite and zircon grains were imaged using backscatter electron (BSE) atomic contrast imaging and cathodoluminescence (CL) imaging; images can be found in Supplementary Figs. 1–4. All scanning electron microscope (SEM) analyses were undertaken on the Tescan Mira3 field emission-gun (FEG) SEM at the Electron Microscopy Facility, within the JdLC. BSE photomicrographs were collected using an accelerating voltage of 15 kV, and CL images were collected with an accelerating voltage of 10 kV.

**Electron backscatter diffraction microstructural analyses.** Shock-deformed monazite and zircon grains were mapped by electron backscatter diffraction (EBSD). Electron backscatter patterns (EBSPs) were collected from the monazite and zircon in orthogonal grids using a Nordlys Nano high-resolution detector and Oxford Instruments Aztec 2.4 acquisition software package on the Mira3 FEG-SEM. EBSD analyses were collected with a 20 kV accelerating voltage, 70° sample tilt, ~20 mm working distance and 18 nA beam current. EBSPs were collected with the following parameters; an acquisition speed of ~40 Hz, 64 frames were collected for a background noise subtraction, 4 × 4 binning, high gain, a Hough resolution of 60 and band detection min/max of 6/8. Maps were collected with a step size between 1.0 and 0.12 μm. Mean angular deviation values of the electron backscatter patterns for the maps ranged between 0.81 and 0.29. Individual zircon grains were mapped using the match unit Zircon 5260 based on the unit cell parameters of Hazen et al.[58] after the methods of Reddy et al.[59]. Monazite grains were mapped with the match unit described in ref. [60], which originates from crystallographic data of ref. [61]. From the Yarrabubba monzogranite (14YB07) seven zircon and four monazite grains were analysed, while eight zircon and seven monazite grains were analysed from the Barlangi granophyre (14YB03).

Post-processing the EBSD data was undertaken with Oxford Instruments Channel 5.11 software suite. All EBSD data were given a wild-spike noise reduction and a six nearest neighbour zero-solution correction. EBSD maps were produced using the Tango suite of Channel 5, while pole figures were processed in the Mambo suite of Channel5. EBSD maps and pole figures (as equal area, lower hemisphere projections) of the shocked monazite and zircons can be found in Supplementary Figs. 1–4. Using Tango the following maps were produced for the shocked zircon and monazite grains:

(1) Inverse pole figure (IPF) maps of crystallographic orientations of zircon (Fig. 2a, c).
(2) All Euler crystallographic orientation map of shocked monazite (Fig. 2b, d).
(3) Grain misorientation map, using the grain rotation orientation direction (GROD)-angle function of Channel5, which helps visualise the substructure of the grains by plotting the deviation angle of each pixel from the mean grain orientation, grain boundaries are defined as >10°. Blue domains are low strains, while warm colours represent higher degrees of misorientation (Supplementary Figs. 1–4).

**Secondary ion mass spectrometry U–Pb age analyses.** Following EBSD mapping of monazite and zircon, in situ U–Th –Pb isotopic measurements targeting specific shock microstructural domains were carried out using the SHRIMP-II secondary ion mass spectrometer (SIMS) at the JdLC. Operating procedures for uranium, thorium and lead isotopic measurements on zircon are based on those described by Compston et al.[62] and Claoué-Long et al.[63], with modifications summarised by Williams[64]. SHRIMP U–Pb zircon and monazite data are reduced using SQUID 2.50 and Isoplot 3.71 (add-ins for Microsoft Excel[65,66]) with decay constants recommended by Steiger and Jäger[67]. Ratios of $^{206}Pb+/^{238}U+$ in zircon are calibrated to the known $^{206}Pb/^{238}U$ of the zircon standard, using a power-law relationship between $^{206}Pb+/^{238}U+$ and $UO^+/U^+$, with a fixed exponent of 2.0 (determined empirically from measurements of zircon standards over several years[63]). All zircon analyses were run during one session and were standardised with primary zircon reference material BR266 (559 Ma, $^{206}Pb/^{238}U = 0.09059$ (ref. [68])) and concentration reference Temora (416.8 Ma, $^{206}Pb/^{238}U = 0.06683$ (ref. [69])). Archaean zircon reference OGC (3465 Ma, $^{207}Pb/^{206}Pb = 0.29907$ (ref. [70])) was analysed during the session to check for $^{207}Pb/^{206}Pb$ fractionation. No $^{207}Pb/^{206}Pb$ fractionation correction was deemed necessary. Eight analyses of the BR266 standard were obtained during the session which indicated an external spot-to-spot (reproducibility) uncertainty of 1.42% (1σ) and a $^{238}U/^{206}Pb*$ calibration uncertainty of 0.54% (1σ). Calibration uncertainties are included in the errors of $^{238}U/^{206}Pb*$ ratios and dates listed in Supplementary Table 1. Common Pb corrections were applied to all analyses using contemporaneous isotopic compositions determined according to the model of Stacey and Kramers[71].

Detailed SHRIMP operating procedures for monazite are outlined in Foster et al.[72] and Wingate and Kirkland[73]. A ~10 μm diameter primary beam was employed with an intensity of ~0.5 nA. Ion microprobe analyses of monazite are affected by an uneven background spectrum of scattered ions[74], which can be reduced effectively by use of the SHRIMP retardation lens system, which is set at ~10 kV. This discriminates against low-energy ions entering the collector. Each analysis consists of six cycles through the isotopic masses in the following sequence: 202 (species $[^{139}La^{31}P^{16}O_2]+$, count time 2 s), 203 ($[^{140}Ce^{31}P^{16}O_2]+$, 2 s), 204 ($^{204}Pb+$, 10 s), 204.1 (background, 10 s), 206 ($^{206}Pb+$, 10 s), 207 ($^{207}Pb+$, 30 s), 208 ($^{208}Pb+$, 5 s), 232 ($^{232}Th+$, 5 s), 254 ($[^{238}U^{16}O_2]+$, 5 s), 264 ($[^{232}Th^{16}O_2]+$, 2 s) and 270 ($[^{238}U^{16}O_2]+$, 3 s). The monazite standard "India"

was used for concentration calibration (509 Ma 2890 ppm $^{238}U$[74]) and also U–Pb calibration. Ratios of $^{206}Pb+/^{238}U+$ in monazite are calibrated to the known $^{206}Pb/^{238}U$ of the monazite standard using a linear relationship between $^{206}Pb+/UO_2+$ and $UO+/UO_2+$[74]. Monazite generates an unresolvable isobaric interference on $^{204}Pb+$, which may be $(^{232}Th^{144}Nd^{16}O_2)++$[75,76]. This interference has been observed to correlate with thorium content[74]. Excess $^{204}Pb+$ counts are corrected against the India monazite standard assuming $^{206}Pb/^{238}U–^{207}Pb/^{235}U$ age-concordance of the standard at a known thorium concentration. Fractionation of the $^{207}Pb/^{206}Pb$ ratio is typically observed when the retardation lens system is at operating voltage during monazite analysis. Fractionation of the $^{207}Pb/^{206}Pb$ ratio is monitored and corrections were applied, if necessary, by reference to the GM3 monazite standard[77] which was run as an unknown. Uncertainties associated with this correction are added in quadrature to the uncertainties of $^{207}Pb*/^{206}Pb*$ ratios and dates. The common-Pb correction was based on measured $^{204}Pb$ and Stacey and Kramers[71] crustal Pb composition appropriate for the age of the sample. Data were reduced using SQUID 2 software and plotted using Isoplot 3.66 (ref. [66]).

Nine SHRIMP analyses from 7 zircon grains and 16 SHRIMP analyses from four monazite grains were collected from the sample of the Yarrabubba monzogranite 14YB07. Nineteen SHIRMP analyses from 8 zircon grains and 19 SHRIMP analyses from 7 monazite grains were collected from the sample of the Barlangi granophyre 14YB03. Selection criteria for the analytical spots were based on the EBSD data, and both the strained domains and the strain-free neoblastic domains of the shocked monazite and zircon were targeted.

U–Pb isotopic data are provided in Supplementary Tables 1 and 2; uncertainties given for individual analyses in the tables (ratios and ages) are at the 1σ level. Terra–Wasserburg concordia plots with 2σ error ellipses for all analyses are shown in Fig. 2. Age uncertainties cited in the text are at the 2σ level. SHRIMP analytical pit locations are documented in Supplementary Figs. 1–4.

**Hydrocode impact simulations.** To better constrain the environmental effects of a Yarrabubba-sized impact structure into a continental ice sheet, the formation of various impact craters were computed using the iSALE shock physics code[46–48], which is based on the original SALE algorithm[78]. The initial conditions were a granitic target covered by an ice sheet of varying thickness (2–5 km). For all ice thicknesses considered here, an impactor diameter of 7 km produced a final crater of ~70 km diameter. The impactor was prescribed an impact velocity of 17 km/s, the current best estimate for the average speed of bolides striking Earth. The ANEOS equation of state for granite[79] was used to model both the impactor and the granitic target and the 5-phase SESAME equation of state for ice[80] was used to model the overlying ice sheet. A computational cell size of 125 m was used, resulting in a resolution of 28 cells per projectile radius. The end product was an impact crater with a ~70 km diameter impact structure, with a zone 40–50 km in diameter in the granite that experienced high shock pressures (>50 GPa) and is heavily fractured, which in some simulations produces an uplifted region. The mass of ice that vapourised during the impact was determined by comparing the entropy of Lagrangian tracer particles in the ice sheet to the critical entropy for incipient (entropy required to go to liquid and vapour state) and complete vaporisation (entropy required to go to a completely vapour state) for ice. In the model with a 2 km thick ice sheet, $1.0 × 10^{15}$ kg of $H_2O$ was above the critical entropy for incipient vaporisation, and $8.7 × 10^{13}$ kg of $H_2O$ was completely vapourised. The mass of ice vapourised in each scenario with different ice thicknesses can be seen in Fig. 5d. Snapshots of the iSALE model at time 0, 36 and 780 s can be found in Fig. 5.

## Data availability

All data used in this manuscript are included in the Supplementary Information file. For access to geologic specimens contact author T.M.E.

## Code availability

The Isoplot and Squid programs used for U–Pb data processing are available through the Berkley Geochronology Center http://www.bgc.org. iSALE is a widely used shock physics code distributed to registered academic non-commercial users via a GitHub repository. Scientists interested in using or developing iSALE should see http://www.isale-code.de for a description of application requirements.

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

## Acknowledgements

T.M.E. would like to thank the Geological Society of Australia, Western Australia Division, for supporting fieldwork at Yarrabubba. E. Thern is thanked for assistance during fieldwork, and D. Kring and M. Schmieder are thanked for fruitful conversations. The ARC, Curtin University, University of Western Australia and CSIRO are acknowledged for funding the Tescan Mira3 FEG-SEM (ARC LE130100053) and the SHRIMP-II housed in the John De Laeter Centre, Curtin University. We thank the developers of iSALE (isale-code.de). T.M.D. was supported by STFC grant ST/S000615/1.

## Author contributions

T.M.E. designed and undertook all experiments, N.E.T. and A.J.C. assisted with EBSD data processing, C.L.K. collected and processed all SHRIMP data, and T.M.D. undertook all iSALE impact modelling simulations. All authors were involved in the production of this manuscript.

## Competing interests

The authors declare no competing interests.
