## [Peer Review File · Nature Communications]

Reviewers' comments:

Reviewer #1 (Remarks to the Author):

Erickson et al. provide an exciting new date on the Yarrabubba impact structure. The techniques are sound and the date is robust establishing it as the oldest preserved impact structure on Earth. This in itself is an excellent discovery and worth of publishing at Nature Comm. Unfortunately, the authors combine this work with a lot of unnecessary speculation about the Paleoproterozoic Snowball. If Yarrabubba was originally ~70km in diameter, we know of ~10 other impacts that were that size or larger, and some of these like the Manicouagan and the Morokweng had no discernable environmental effects. Although the idea that this impact ended the Paleoproterozoic Snowball is interesting, relating this impact structure to the Great Oxidation event is a bit much, unless they want to suggest this was part of a cluster. There is also a lot of commentary in the text that is stated as fact but is instead uncertain—e.g. the cause of glaciation is unknown yet the authors just state 'global icehouse conditions resulted from drawdown of atmospheric greenhouse gases by early life during the Great Oxidation Event'—this is total speculation and not relevant to the contribution. Basically, the whole section 'Implications for the Paleoproterozoic Earth' needs to be deleted (particularly lines 143-156) and completely rewritten. This section is just distraction and not relevant to the data. If the authors just cut some of the speculation, I think this will be an excellent contribution.

In regards to the age of the structure, and explained below, in our 2003 EPSL paper, in section 5 (Age), we said "One zircon from the granophyre gave a concordant Paleoproterozoic age [14], which is unusual for igneous rocks from the northern Yilgarn. It is tempting to suggest that this is the age of impact." Fletcher's 2234 age is, of course, within error of the new Erickson results, so our suggestion was almost certainly correct. In this respect, Macdonald et al. and Fletcher could be cited more directly as suggesting this age. A 2000+ age for Yarrabubba also makes sense because deep erosion of the structure would have to have happened when the Yilgarn was eroded to a peneplane before the early Proterozoic transgressions along the northern margin of the Yilgarn at around 1900 Ma. The authors could include this point as supporting data.

Line comments:

2: Replace 'astrobleme' with 'impact structure'—most people don't know what an astrobleme is.

18: Cut 'Great Oxidation Event'—not relevant.

25-26: Cut all of this. Just state that 'Within analytical uncertainty, the Yarrabubba impact event coincided with the end of the Gowganda-Reitfontein glaciation.'

55-56: Rewrite. 'released significant water and greenhouse gases that may have triggered deglaciation'

58: delete 'nominally'—we showed that it was cut by Proterozoic dykes and Piranjo showed that it had Proterozoic hydrothermal alteration

80-81: Ian Fletcher got 2650 Ma (emplacement) and 2700-2790 Ma (xenocryst) U-Pb zircon SHRIMP ages from the granite, and 2715 Ma (xenocryst) ages from the granophyre, except for one zircon from the granophyre that gave 2234 Ma. The younger zircon was originally ignored as an unexplained outlier, and Fletcher didn't want us to publish the 2234 age. In our EPSL paper, in section 5 (Age), we said "One zircon from the granophyre gave a concordant Paleoproterozoic age [14], which is unusual for igneous rocks from the northern Yilgarn. It is tempting to suggest that this is the age of impact." Fletcher's 2234 age is, of course, within error of the new Erickson results, so our suggestion was almost certainly correct. In this respect, Macdonald et al. and Fletcher could be cited more directly as suggesting this age.

143-144: Delete. 2234 Ma post-dates GOE.

144: Delete. This is just a transition of the MIF signal in Sulfur, not a transition from anoxic to aerobic.

145-147: This model for initiation is stated as fact, but is not accepted at all. Instead initiation may have had nothing to do with life but instead global weatherability and CO₂ consumption.

147-149: Rewrite. These are not commonly referred to as the Huronian snowball Earth event, but

rather, the 2234 Ma age coincides with the end of the third of three glacial intervals, which is better referred to as the Gowganda-Reitfontein glaciation.
149-152: Delete. These are not Phanerozoic-style glacial-interglacial cycles driven by Milankovich. These are glacial and non-glacial climate states. Do not use the term 'interglacial cycles' for 10s of millions of year intervals. Moreover, what does any of this have to do with the impact?
154: What glacial deposits are the authors referring to at ~1.63 Ga?
154-156: Delete. This is nonsense. Evidence for a lull in tectonic activity is totally speculative and again not appropriate for this paper.
158: Replace 'Huronian' with 'Paleoproterozoic glaciations'
161: Not clear—the age of the Lomagundi excursions are still poorly constrained.
163-164: Replace 'a tipping point in an already dynamic period of global climate change' with 'deglaciation'.
180: Replace 'Huronian' with 'Paleoproterozoic glaciations'
184: Replace 'Huronian' with 'Paleoproterozoic glaciations'
185-186: If you are going to end with a speculative statement, I would instead suggest that "the Yarrabubba impact may have ended glaciation and saved life from a longer, more severe Snowball."

Francis Macdonald

Reviewer #2 (Remarks to the Author):

See attached pdf for reviewer's report by P.F. Hoffman (24 April 2019).

Reviewer #3 (Remarks to the Author):

The MS describes U-Pb dating of shocked minerals from Yarrabubba impact structure and reports that it is the oldest partially preserved impact structure on Earth which is ~200 millions years older than the oldest known so far Vredefort crater.

I am not an expert in U-Pb isotope chemistry and cannot evaluate this part of the MS (which is the most important to my opinion). Hopefully, other reviewers can do it. The numerical model of an impact into ice sheet with a thickness from 2 up to 7 km looks reliable. However, I have some concerns about the conclusion that ejection of a vast amount of water vapor could be a trigger for a global climate change:

1. The total mass of water vapor ejected into the atmosphere may be different from the total amount of vaporized (subjected to high enough shock pressure) ice, i.e., not all shocked ice is ejected from the crater.

2. Liquid water (which is part of the incipiently vaporized ice) is removed from the atmosphere quickly and has no any climate effects.

3. Water vapor ejected into the troposphere is removed quickly and has no climate effects.

4. The atmosphere cannot hold more than 10^{15} kg of water vapor (e.g., Pierazzo 2005 and references therein).

4. The net effect of vapor/dust ejection could be much more complicated than simple global warming and requires to run a climate model which is certainly beyond the goals of this MS.

In summary, I would recommend to make the last section (Implications for the Palaeoproterozoic Earth) a bit more modest – in contrast to the rest of the MS the hypothesis is quite speculative.

It will be also interesting to see how the total amount of granitic melt changes with an increase of the ice sheet thickness – are their enough melt to produce the magnetic anomaly?)

In addition,

1. I would recommend to change the title – to substitute "astrobleme" by "impact structure" and

re-phrase. An "astrobleme" is a romantic term which would be better to avoid in scientific papers.

2. Lines:34-35. As the authors discussed the oldest structures which are <3 Gyr old, I would not state that the bombardment rate was orders of magnitude larger - it is practically constant during the last 3 Gyr and was probably only slightly higher at 3.47 Gyr (the oldest known impact products on Earth).

3. Line 32. "Meteorite" better substitute by extraterrestrial. By definition, meteorites are fragments of extraterrestrial materials which keep their initial structure (i.e., are not vaporized/melted upon impact).

Reviewer #1 (Remarks to the Author):

Erickson et al. provide an exciting new date on the Yarrabubba impact structure. The techniques are sound and the date is robust establishing it as the oldest preserved impact structure on Earth. This in itself is an excellent discovery and worth of publishing at Nature Comm. Unfortunately, the authors combine this work with a lot of unnecessary speculation about the Paleoproterozoic Snowball. If Yarrabubba was originally ~70km in diameter, we know of ~10 other impacts that were that size or larger, and some of these like the Manicouagan and the Morokweng had no discernable environmental effects. Although the idea that this impact ended the Paleoproterozoic Snowball is interesting, relating this impact structure to the Great Oxidation event is a bit much, unless they want to suggest this was part of a cluster. There is also a lot of commentary in the text that is stated as fact but is instead uncertain—e.g. the cause of glaciation is unknown yet the authors just state ‘global icehouse conditions resulted from drawdown of atmospheric greenhouse gases by early life during the Great Oxidation Event’—this is total speculation and not relevant to the contribution. Basically, the whole section ‘Implications for the Paleoproterozoic Earth’ needs to be deleted (particularly lines 143-156) and completely rewritten. This section is just distraction and not relevant to the data. If the authors just cut some of the speculation, I think this will be an excellent contribution.

The effects of impacts events on ambient global environmental conditions are complex and thus may or may not result in a large-scale changes to Snowball like environments (e.g. Ivanov and Koeberl, 2019 MAPS). We consider the contemporaneity of Yarrabubba with the termination of the youngest Palaeoproterozoic glacial deposit, the Rietfontein of Southern Africa, and the initiation of a significant shift to strongly positive carbon isotope values in the global geological record to be extraordinary enough to simply highlight in our paper. It follows that the potential mechanisms for any causative relationships are also discussed. Nevertheless, we have removed reference to the Great Oxidation Event. We have also modified sections on the potential causes of the atmospheric cooling that initiated Palaeoproterozoic snowball Earth. The final section has also been modified significantly to highlight the coincidence of Yarrabubba with the end of Palaeoproterozoic diamictite deposits, discuss the known climatic effects of large impacts, show modelling results for a Yarrabubba sized impact into a continental ice sheet and discuss potential climate effects of Yarrabubba more judiciously and work is focused on the new age data.

In regards to the age of the structure, and explained below, in our 2003 EPSL paper, in section 5 (Age), we said “One zircon from the granophyre gave a concordant Paleoproterozoic age [14], which is unusual for igneous rocks from the northern Yilgarn. It is tempting to suggest that this is the age of impact.” Fletcher’s 2234 age is, of course, within error of the new Erickson results, so our suggestion was almost certainly correct. In this respect, Macdonald et al. and Fletcher could be cited more directly as suggesting this age. A 2000+ age for Yarrabubba also makes sense because deep erosion of the structure would have to have happened when the Yilgarn was eroded to a peneplane before the early Proterozoic transgressions along the northern margin of the Yilgarn at around 1900 Ma. The authors could include this point as supporting data.

We acknowledge the contribution of Macdonald et al. (2003) to setting the foundation for this study. As such we have stated that the zircon age is ‘within uncertainty of a single previously reported U – Pb

zircon analysis from the Barlangi granophyre of 2234 ± 28 Ma²⁷, which was inferred by²¹ to indicate a Palaeoproterozoic impact age.’ Line 121-123.

Line comments:

2: Replace ‘astrobleme’ with ‘impact structure’—most people don’t know what an astrobleme is.

The title has been modified to remove ‘astrobleme’.

18: Cut ‘Great Oxidation Event’—not relevant.

Done.

25-26: Cut all of this. Just state that ‘Within analytical uncertainty, the Yarrabubba impact event coincided with the end of the Gowganda-Reitfontein glaciation.’

This sentence has been rephrased to remove speculation and reference to the GOE.

55-56: Rewrite. ‘released significant water and greenhouse gases that may have triggered deglaciation’

We have made the suggested edit.

58: delete ‘nominally’—we showed that it was cut by Proterozoic dykes and Piranjo showed that it had Proterozoic hydrothermal alteration

Done.

80-81: Ian Fletcher got 2650 Ma (emplacement) and 2700-2790 Ma (xenocryst) U-Pb zircon SHRIMP ages from the granite, and 2715 Ma (xenocryst) ages from the granophyre, except for one zircon from the granophyre that gave 2234 Ma. The younger zircon was originally ignored as an unexplained outlier, and Fletcher didn’t want us to publish the 2234 age. In our EPSL paper, in section 5 (Age), we said “One zircon from the granophyre gave a concordant Paleoproterozoic age [14], which is unusual for igneous rocks from the northern Yilgarn. It is tempting to suggest that this is the age of impact.” Fletcher’s 2234 age is, of course, within error of the new Erickson results, so our suggestion was almost certainly correct. In this respect, Macdonald et al. and Fletcher could be cited more directly as suggesting this age.

We have now integrated the age suggestion of Macdonald et al. (2003) in lines 122-123. However, we would like to note that the anomalous zircon age of Fletcher & McNaughton of 2234 Ma was not discussed at all by the authors who did this dating. As stated above Macdonald et al. (2003) specified “One zircon from the granophyre gave a concordant Paleoproterozoic age [14], which is unusual for igneous rocks from the northern Yilgarn. It is tempting to suggest that this is the age of impact, but more zircons need to be analyzed before the significance of the Paleoproterozoic age can be established.” Macdonald et al. (2003) hence did not directly provide a precise age but rather speculated that the impact occurred during the “Paleoproterozoic” eon. The authors also provide an argument for undertaking more geochronology to determine the impact age with certainty, which we have done in this study. In

any case, we do very much recognize the contribution of Macdonald et al. (2003) in the revised text to correctly placing Yarrabubba into its temporal context.

143-144: Delete. 2234 Ma post-dates GOE.

We have removed the GOE from this sentence.

144: Delete. This is just a transition of the MIF signal in Sulfur, not a transition from anoxic to aerobic.

This has been deleted.

145-147: This model for initiation is stated as fact, but is not accepted at all. Instead initiation may have had nothing to do with life but instead global weatherability and CO₂ consumption.

We have removed discussion of the mechanisms which transitioned Earth into snowball conditions during the Palaeoproterozoic.

147-149: Rewrite. These are not commonly referred to as the Huronian snowball Earth event, but rather, the 2234 Ma age coincides with the end of the third of three glacial intervals, which is better referred to as the Gowganda-Reitfontein glaciation.

We have replaced 'Huronian snowball Earth' with 'Palaeoproterozoic snowball Earth' throughout this section.

149-152: Delete. These are not Phanerozoic-style glacial-interglacial cycles driven by Milankovich. These are glacial and non-glacial climate states. Do not use the term 'interglacial cycles' for 10s of millions of year intervals. Moreover, what does any of this have to do with the impact?

We have modified this to removed reference to the glacial-interglacial cycles and deleted this sentence.

154: What glacial deposits are the authors referring to at ~1.63 Ga?

We thank the reviewer for catching this error and have fixed this with >400 Ma after Williams (2005, Journal of the Geological Society), which documented 1.8 Ga low-latitude glacial markers in the Kimberley, Western Australia.

154-156: Delete. This is nonsense. Evidence for a lull in tectonic activity is totally speculative and again not appropriate for this paper.

A published (Nature Geoscience; Spencer et al., 2018) paper based on a comprehensive evaluation of the geologic record from 2.4 to 2.0 Ga indicates that the frequency of magmatic rock production, ages of orogenic belts, passive margins and large igneous provinces diminished in this time period. Collectively, this large data set was used to argue for a reduction in tectono-magmatic activity in this specific time period (Spencer et al., 2018). This concept is widely accepted, 13 citations in a year for

Spencer et al., (2018). Hence, we feel it appropriate to discuss this aspect given the end of the Palaeoproterozoic snowball Earth coincides with this apparent tectonic lull and thus the termination of snowball Earth appears on the basis of this data to be incompatible with tectonic drivers, highlighting the need to discuss other mechanisms such as impact cratering in the contribution.

158: Replace 'Huronian' with 'Paleoproterozoic glaciations'

We have removed 'Huronian' throughout this section and replaced it with 'Palaeoproterozoic glaciations'.

161: Not clear—the age of the Lomagundi excursions are still poorly constrained.

We have included reference to Martins et al., which provides a review of the geochronological constraints on Lomagundi event. The best constrained age is between 2221 ± 5 Ma and 2106 ± 8 Ma, providing evidence for contemporaneous onset of the Lomagundi excursion with the Yarrabubba impact event.

163-164: Replace 'a tipping point in an already dynamic period of global climate change' with 'deglaciation'.

We have modified this section.

180: Replace 'Huronian' with 'Paleoproterozoic glaciations'

Done.

184: Replace 'Huronian' with 'Paleoproterozoic glaciations'

Done.

185-186: If you are going to end with a speculative statement, I would instead suggest that "the Yarrabubba impact may have ended glaciation and saved life from a longer, more severe Snowball."

We have completely rewritten the final few sentences of the conclusions to try to remove such a 'speculative statement'.

Francis Macdonald

Summary

The reliable age reported here for the Yarrabubba impact structure, making it the world's oldest astrobleme, is newsworthy and suitable for publication in *Nature Communications*. However, one aspect of the paper requires minor revision and two topics need major revision. The minor revision concerns previous estimates of the age of the impact structure. Major revisions are needed in the characterization of the geologic record of Paleoproterozoic glacial history, which contains many factual errors and misleading statements. A snowball earth climate state is assumed in the ms. to have existed at the time of the impact, but in fact the sole reliable geological (paleomagnetic) evidence for a Paleoproterozoic snowball state pertains strictly to a glaciation that occurred 200 million years earlier. The climate at the time of the impact could have been nonglacial, glacial (like present) or panglacial: we just don't know. Major revisions are also needed in the speculations on possible climatic consequences of the impact event. These appear naïve and superficial. In particular, the atmospheric lifetime of the water vapour generated by the impact (a small fraction of the ambient ice in a snowball state) needs to be considered. If the water vapour rapidly re-condenses and settles out, on a timescale of weeks to years, its quasi-greenhouse effect on climate will be severely limited by its duration, given the enormous heat sink of the ambient ice. As for surface albedo effects, airborne dust particles are likely to become nuclei for ice crystals, in which case their albedo effect as fallen snow may be subtle, except in the equatorial ablation zone (see Pierrehumbert et al., 2011, *Annu. Rev. Earth Planet. Sci. Lett.* 39, 417-460) where dust accumulation in excess of ambient would temporarily lower the surface albedo. My recommendation is that the paleoclimatic implications (p. 7-9) be either greatly downplayed, or brought up to scientific standard in terms of the geology and climatology. In short, stay impact or get serious.

Specific comments by line number

19-20 Actually, ref. 20 (Macdonald et al., 2003) favoured a "Paleoproterozoic" age for the Yarrabubba impact, citing the 2234 ± 34 Ma U-Pb age²⁶ for the Berlangi granophyre quoted in Cassidy et al. (2002). They also noted that the impact should have occurred before the peneplanation preceding sedimentary onlap (Nabberu basin) from the north around 1.9 Ga. It would do the present authors no harm to cite the prescient, and justly tentative, age estimate of Macdonald et al. (2002), whose paper which established the impact origin for the structure is no less creditable coming from one who, at the time, held a B.Sc. degree only, yet was considered worthy enough to inherit the late Gene Shoemaker's impact survey of Australia.

We entirely agree and have incorporated the tentative age inferred by Macdonald et al. (2003) into the text (Lines 122-123). Please see the above discussion of the initial age estimate of Macdonald et al. (2003) for further details around this aspect.

53-54 There are three discrete glacial episodes in the Huronian Supergroup, and they are only known to have occurred somewhere between 2.45 and 2.31 Ga.

The 2.31 Ga age is from a “microtuff” in the Gordon Lake Formation, which lies 1.0–1.6 km stratigraphically *above* the top of the youngest glacial unit (Gowganda Formation). In all likelihood, the Gowganda glaciation ended tens of millions of years before 2.31 Ga.

We thank the reviewer for highlighting glacial episodes recorded by the Huronian Supergroup. However, our focus is the final episode of Palaeoproterozoic Snowball Earth (i.e., spanning multiple cratons), which we erroneously termed the Huronian (a term which the reviewer asserts should be limited to the Superior craton) Snowball Earth. We have corrected our error in the terminology throughout the manuscript, and focused this section to discuss contemporaneity between the Reitfontein glacial termination and the Yarrabubba impact event more explicitly.

55–56 In a snowball climate, indeed in any climate, the impact-generated water vapour would rapidly condense/precipitate as rain or snow in the cold lower atmosphere. As cloud condensate they would still have a quasi-greenhouse effect until they settle as rain or snow, but the warming might be negated by cloud albedo effects if the ambient climate was not a snowball. Note that Bendtsen & Bjerrum (2002, *Geophys. Res. Lett.* **29**(15), 10.1029/2002GL014829) found that a somewhat larger impact than Yarrabubba could *trigger* a snowball climate if the ambient ocean was as cold as present, not warm like the Late Cretaceous.

The reviewer raises an interesting set of points on the effects of impact-generated vapour plumes on the lower atmosphere. However, one of the important aspects of the vapour plume produced by an impact event is that the water vapour liberated will reach the upper atmosphere and even interact with the atmosphere from above as it is gravitationally drawn back to Earth (e.g. Koeberl and Ivanov, 2019, MAPS). Thus, the interactions of the shock vapourized water with the Earth’s atmosphere are not limited solely to the lower atmosphere, and may have more complex interactions with and widespread effects upon the upper atmosphere. Consequently, the atmospheric effects may not be as shortlived as the reviewer suggests. We have attempted to make this point clearly in the manuscript.

145–147 That CO₂ drawdown during the GOE was a consequence of “the evolution and proliferation of photosynthesizing organisms” is an outlier hypothesis held by Kirschvink and Kopp. The consensus view is that oxygenic photosynthesis arose hundreds of Myr before the GOE. A more common hypothesis is that the rise of O₂ (and OH) itself led to reduced greenhouse warming, through oxidative destruction of reduced GHG species. This hypothesis, also, lacks empirical support.

We have removed discussion of the cause of the CO₂ drawdown that has been proposed to have initiated Palaeoproterozoic snowball Earth.

147 The term “icehouse” with reference to global climate is an abomination. An icehouse is an enclosure that is kept cold by placing blocks of ice inside. Global climate doesn’t work this way: the heat absorbed by the blocks of

ice is the same heat that was given off when the blocks were frozen to begin with. When you feel cold standing next to a glacier, that's an "icehouse" effect. When ice sheets appear where they were formally non-existent, that's something else.

Ok, the term icehouse has been replaced by 'snowball Earth episode'.

148-149 "Huronian snowball Earth event": There is no credible evidence for low-latitude or snowball glaciation in the Huronian Supergroup. Low-latitude paleomagnetic poles obtained from Huronian rocks overlying the glaciogenic Gogwanda Formation (Williams & Schmidt, 1997, *Earth Planet. Sci. Lett.* **153**, 273-285; Schmidt & Williams, 1999, *Earth Planet. Sci. Lett.* **172**, 271-285) have been shown to be post-depositional overprints (Hilburn et al., 2005, *Earth Planet. Sci. Lett.* **232**, 315-332). One of the older Huronian glaciations could well be correlative with the low-latitude ($11\pm 6^\circ$) Makganyene glaciation (Evans et al., 1997, *Nature* **386**, 262-266) in South Africa, but in this case the glaciation should be referred to as Makganyene not Huronian. Deposition of the upper Makganyene Formation coincided with eruption of the Ongeluk volcanics, recently dated at 2426 ± 3 Ma³⁸. Accordingly, the Makganyene snowball terminated as much as 200 Myr before the Yarrabubba impact.

As stated above we have removed reference to 'Huronian snowball Earth event' and now focus on the age of the Rietfontein glaciation and subsequent > 400 Ma lull in diamictite deposits after the Yarrabubba impact event.

152-153 Huronian glaciation did not terminate at ca. 2230 Ma as stated here, the last Huronian glaciation ended well before 2.31 Ga (Bekker et al., 2010, *AGU Fall Mtg Abs.* U32A-06). A kilometer-thick regional-scale sandstone sheet (Lorrain Formation) was deposited after the last Huronian glaciation and before the 2.31 Ga tuff horizon, and slow tectonic subsidence was required to accommodate that sandstone sheet.

We have removed reference to the Huronian glaciation, which we hope removes confusion / speculation based solely on the glacial deposits of the Superior craton.

157-158 The Yarrabubba age demonstrates that the impact postdates any Huronian glaciation by at least 80 Myr (2.23 vs 2.31 Ga) and more likely by >100 Myr.

As stated above we have removed reference to the Huronian glaciation, focusing on the Rietfontein deposit from the Kaapvall Craton in South Africa due to its significance as representing the final glacial episode of Palaeoproterozoic snowball Earth.

160 It is possible that the Yarrabubba impact coincided with termination (or onset) of the Rietfontein glaciation in South Africa. However, the 2225 ± 3 Ma age cited is only a minimum age constraint on Rietfontein glaciation, not the age of glacial termination. The age (mis-cited in the ms., being actually from D.H. Dorland's 2004 PhD thesis at U. Johannesburg) is a TIMS U-Pb date of a single, non-abraded, zircon grain and could, despite its concordance, be too young if it suffered early Pb loss. As an age for Hekpoort volcanism, it should not be too heavily relied upon.

We hope that correcting the nomenclature issue with the Palaeoproterozoic snowball Earth the issues in contemporaneity have been resolved. We have also corrected the mis-citation issue with H. C Dorland's thesis, and noted that it is a 'minimum age constraint'.

162 For age constraints on the Lomagundi positive carbon isotope interval, add reference to Martin *et al.* (2013, *Earth-Sci. Rev.* **127**, 242-261).

We have integrated the summary of age constraints for the Lomagundi excursion from Martin *et al.* 2013 into the manuscript.

173-174 The global and regional (Yilgarn) climate state at 2.23 Ga is unknown. The Yilgarn could have been ice covered or ice free. The global climate could have been nonglacial, glacial or panglacial. The climatic consequence of the impact could be quite different, depending on the ambient climate state.

This is an excellent point and has been bolstered within the final section of the manuscript, we currently do not know the specific nonglacial, glacial or panglacial state of the Yilgarn at this time period but do point to the general planetary state at this time. Age constraints on the Meteoritebore member diamictite of the Hamersley province to the north of the Yilgarn, are poor.

177-180 The estimated mass of water vapour released upon impact (9×10^{13} to 5×10^{15} kg) pales beside the estimated 2×10^{17} kg of combined marine and continental ambient ice on a snowball earth (Abbott *et al.*, 2013, *J. Geophys. Res. Atmos.* **118**, 6017-6027; Benn *et al.*, 2015, *Nature Geosci.* **8**, 704-708). This ice represents an enormous heat sink, and to melt it would require 5–6 W m² of greenhouse radiative forcing *sustained over 2-3 kyr* (Wallace & Hobbs, 1977, *Atmospheric Science: An Introductory Survey*. Academic Press, p. 320).

We thank the reviewer for bringing this key point to our attention and have integrated the mass of water locked up as ice during a snowball period, and thus the enormity of a heat sink that this would cause (Line 185-188).

180 Why the comparison with the stratosphere, which contains only a tiny fraction of the water vapour content of the lower troposphere? The appropriate comparison is the water vapour content of the upper troposphere before and after the impact, since that is where the

radiative balance and hence the greenhouse effect actually operate.

We have removed the comparison with stratospheric water load. Although it should be noted the vapour plume likely transported large quantities of water vapour into the upper atmosphere, and therefore potentially deposited water vapour into the lower atmosphere from above.

181 The hydrologic cycle does not “shut down” in a snowball earth, but the atmosphere *is* very dry because of the cold air (saturation vapour pressure scales with temperature, and is closely similar over ice or super-cooled water at any given temperature). Consequently, the hydrologic cycle is weak, but far from inconsequential over the time scale of a snowball episode.

Thank you for this clarification we have amended the statement from ‘shutdown’ to ‘slowdown’.

184-186 The climatic consequences of an impact resulting from greenhouse radiative forcing of released water vapour must critically depend on the duration of the radiative forcing. If the impact-generated water vapour condensed or froze out rapidly in the cold and oversaturated atmosphere, the radiative forcing from water vapour and cloud IR absorption combined would be short-lived. The large mass of ambient ice gives a snowball earth enormous thermal inertia. An estimated $\sim 6.7 \times 10^{22}$ Joules of heat would be required to melt the ice remaining after the impact. One is left skeptical that a Yarrabubba-size impact could terminate a snowball that was not already near termination (in which case the impact would not have changed the course of history). I suspect that the Yarrabubba impact may have had a more significant climate impact if the ambient climate was *not* a snowball (e.g., Bendtsen and Bjerrum, 2002). Your reviewer is not a climate physicist but a field geologist, yet in this day and age it is the duty of all scientists to discuss climate change with circumspection.
–P.F. Hoffman (24 April 2019)

We thank the reviewer for making this excellent point. We have refocused the Palaeoproterozoic implications’ section of the paper with the aim of reducing those more speculative elements. While the starting conditions for the Earth’s atmosphere at the time of the Yarrabubba impact structure are unknown, the potential correlation of the Yarrabubba impact event with the termination of the Rietfontein diamictite and the onset of the Longumdi isotopic excursion are striking. We therefore feel it is reasonable and in fact pertinent to spend a few words highlighting this temporal relationship, and discuss the feasibility of causation given our predictions of ejected water vapour via shock physics simulations and the current understanding of climate forcing effects.

Reviewer #3 (Remarks to the Author):

The MS describes U-Pb dating of shocked minerals from Yarrabubba impact structure and reports that it is the oldest partially preserved impact structure on Earth which is ~200 millions years older than the oldest known so far Vredefort crater.

I am not an expert in U-Pb isotope chemistry and cannot evaluate this part of the MS (which is the most important to my opinion). Hopefully, other reviewers can do it. The numerical model of an impact into ice sheet with a thickness from 2 up to 7 km looks reliable. However, I have some concerns about the conclusion that ejection of a vast amount of water vapor could be a trigger for a global climate change:

1. The total mass of water vapor ejected into the atmosphere may be different from the total amount of vaporized (subjected to high enough shock pressure) ice, i.e., not all shocked ice is ejected from the crater.

We have modified these values to reflect the amount of water that is completely vapourised, and thus will be a part of the vapour plume, from the value of water that experiences incipient vapourisation, which may not. Furthermore, we now strongly focus the manuscript into the dating of the Yarrabubba structure.

2. Liquid water (which is part of the incipiently vaporized ice) is removed from the atmosphere quickly and has no any climate effects.

We agree with the reviewer on this well-made point, and so we have removed the discussion of the incipient vapourised ice volumes and focused on the ice which was completely vapourised and thus within the vapour plume, we take this conservative approach to not over estimate the amount of water interacting with the atmosphere.

3. Water vapor ejected into the troposphere is removed quickly and has no climate effects.

We agree with the reviewer that the residence time of tropospheric water vapour is likely to be shortlived. However, the interaction of the water vapour within the vapour plume and the atmosphere, and ultimately the effects on climate are complicated processes (e.g. Koeberl and Ivanov 2019, MAPS) due to the rapid distribution of water over large areas of the planet.

4. The atmosphere cannot hold more than 10^{15} kg of water vapor (e.g., Pierazzo 2005 and references therein).

We thank the reviewer for pointing this out. However, the vapourised ice only produces $\sim 5 \times 10^{14}$ kg that comprises the vapour plume.

4. The net effect of vapor/dust ejection could be much more complicated than simple global warming and requires to run a climate model which is certainly beyond the goals of this MS.

We agree that it would be nice to see climatic simulations of a Yarrabubba impact event run. However, we agree with the reviewer that this is outside of the scope of this manuscript. The focus of this

manuscript is to establish a precise age for Yarrabubba, highlight its coincidence with the end of Palaeoproterozoic snowball Earth, and hypothetically investigate the effects of a Yarrabubba-sized impact on a continental ice sheet using established codes for shock physics. Running climate models would require careful consideration of the initial model parameters, many of which are highly uncertain for this period of Earth history.

In summary, I would recommend to make the last section (Implications for the Palaeoproterozoic Earth) a bit more modest – in contrast to the rest of the MS the hypothesis is quite speculative.

We agree and have decreased the amount of speculation within this section. Nevertheless, we maintain that the effects that a Yarrabubba-sized impact would have on Earth's climate during the Palaeoproterozoic could be an avenue of investigation that is worth further consideration, especially given the temporal correspondence of Yarrabubba to global environmental change.

It will be also interesting to see how the total amount of granitic melt changes with an increase of the ice sheet thickness – are their enough melt to produce the magnetic anomaly?)

The Yarrabubba impact structure is a deeply eroded impact with volumetrically little preserved impact melt rock, and so has only a small magnetic anomaly associated with impact melt. The larger demagnetized zone within Yarrabubba is a result of shock deformation and large scale brecciation of the basement target rocks. While the reviewer's question is generally interesting to impact cratering scientists, it has no direct bearing on observations that can be made at Yarrabubba, and is not considered further here.

In addition,

1. I would recommend to change the title – to substitute “astrobleme” by “impact structure” and re-phrase. An “astrobleme” is a romantic term which would be better to avoid in scientific papers.

We have removed astrobleme from the title.

2. Lines:34-35. As the authors discussed the oldest structures which are <3 Gyr old, I would not state that the bombardment rate was orders of magnitude larger - it is practically constant during the last 3 Gyr and was probably only slightly higher at 3.47 Gyr (the oldest known impact products on Earth).

We removed ‘orders of magnitude’.

3. Line 32. “Meteorite” better substitute by extraterrestrial. By definition, meteorites are fragments of extraterrestrial materials which keep their initial structure (i.e., are not vaporized/melted upon impact).

Done, changed to ‘extraterrestrial’.

Reviewers' comments:

Reviewer #1 (Remarks to the Author):

The manuscript is much improved, but still could be tightened up.

The title could still use some work--the authors could include that it is the oldest recognized impact structure on Earth, but more importantly the title doesn't mention anything of the impact on ice modeling. Maybe something like: "Geochronology of the 2.2 Ga Yarrabubba impact structure and deglaciation of Paleoproterozoic Snowball Earth".

The abstract mentions the Lomagundi carbon isotope excursion--this should be deleted. There is no mechanism, this is not developed, and is a distraction. Again, the authors lose credibility when they throw in random geological features with no mechanistic relationship. The Lomagundi excursion is a positive isotope interval of tens of meters of stratigraphy likely representing millions of years. Why the authors want to associate an impact with this is beyond me.

Line 146-158 again rambles about random geological features that cannot be associated with impacts--this is a distraction and weakens the manuscript. I don't understand why the authors refuse to stick with their data and the main exciting results. The impact is the oldest preserved and is within error of deglaciation--their modeling suggests a possible mechanism. The rest is totally distracting and weakens the manuscript.

Line 195. Again, delete Lomagundi. It is speculative enough relating it to deglaciation--why do you have to try to relate it to all Earth System change with no mechanism? If you are going to get into the carbon cycle, actually get serious and talk about mechanisms and timescales. Otherwise, this vapid speculation undermines what is otherwise an excellent contribution.

Reviewer #2 (Remarks to the Author):

The last two sentences of the abstract are incorrect and misleading.

In lines 22-24 it is stated that the "impact event coincides within analytical uncertainty [i.e. 2229 +/- 5 Ma] with the termination of . . . snowball conditions and commencement of the . . . carbon isotope excursion." But there is no evidence that the Rietfontein glaciation was a snowball event and its age is only constrained as being older than 2225 +/- 3 Ma. It could be up to 100 million years older. The . . . isotope excursion is only known to begin sometime before 2221 +/- 5 Ma, which is the age of an intrusive igneous body that cuts and must therefore be younger than sediments hosting the excursion. The excursion could have begun up to 85 million years earlier (Martin et al. 2013). Accordingly, the overlapping dates (given analytical uncertainties) are more a coincidence than a correlation. No two of the three events are established as being "coincident within analytical uncertainty," let alone all three. This is an intolerable abuse of radiometric data.

Regarding the final abstract sentence (lines 24-27), the authors have not demonstrated that water vapour release by the impact "provides a viable mechanism for affecting the earth's climate" when the implied climate change is snowball termination. Even the residence time of water vapor in the stratosphere (months to years) is likely too short given the enormous heat sink of a largely glaciated planet. Moreover, ice clouds have albedo as well as quasi-greenhouse effects that are complex and counteracting.

I have not gone over the rest of the revised ms. The abstract is unacceptable to me.

PF Hoffman

Response to reviewers

Reviewer #1 (Remarks to the Author):

The manuscript is much improved, but still could be tightened up.

We thank the reviewer for their previous comments which have helped improve the quality of the manuscript.

The title could still use some work--the authors could include that it is the oldest recognized impact structure on Earth, but more importantly the title doesn't mention anything of the impact on ice modeling. Maybe something like: "Geochronology of the 2.2 Ga Yarrabubba impact structure and deglaciation of Paleoproterozoic Snowball Earth".

We agree that it would be nice to include the fact that Yarrabubba is the oldest recognized impact crater on Earth and that we have produced new modelling results for impacts into continental glaciers, however, we feel that with Nature Communication's strict word limit for titles (15) the current title is appropriate.

The abstract mentions the Lomagundi carbon isotope excursion--this should be deleted. There is no mechanism, this is not developed, and is a distraction. Again, the authors lose credibility when they throw in random geological features with no mechanistic relationship. The Lomagundi excursion is a positive isotope interval of tens of meters of stratigraphy likely representing millions of years. Why the authors want to associate an impact with this is beyond me.

We have removed the mention of the Lomagundi carbon isotope excursion from the abstract.

Line 146-158 again rambles about random geological features that cannot be associated with impacts--this is a distraction and weakens the manuscript. I don't understand why the authors refuse to stick with their data and the main exciting results. The impact is the oldest preserved and is within error of deglaciation--their modeling suggests a possible mechanism. The rest is totally distracting and weakens the manuscript.

We have removed the link to the Lomagundi Carbon isotope excursion and geological features that are not associated with the impact event from this section. We feel that the text now highlights the significance of the age of the Yarrabubba impact event within the general geodynamic framework of the Palaeoproterozoic without adding extra distraction.

Line 195. Again, delete Lomagundi. It is speculative enough relating it to deglaciation--why do you have to try to relate it to all Earth System change with no mechanism? If you are going to get into the carbon cycle, actually get serious and talk about mechanisms and timescales. Otherwise, this vapid speculation undermines what is otherwise an excellent contribution.

We have sought to focus the work on the new robust geochronology and removed the inference about implications for the Lomagundi carbon isotope excursion.

Reviewer #2 (Remarks to the Author):

The last two sentences of the abstract are incorrect and misleading.

In lines 22-24 it is stated that the "impact event coincides within analytical uncertainty [i.e. 2229 +/- 5 Ma] with the termination of . . snowball conditions and commencement of the . . carbon isotope excursion." But there is no evidence that the Rietfontein glaciation was a snowball event and its age is only constrained as being older than 2225 +/- 3 Ma. It could be up to 100 million years older. The . . isotope excursion is only known to begin sometime before 2221 +/- 5 Ma, which is the age of an intrusive igneous body that cuts and must therefore be younger than sediments hosting the excursion. The excursion could have begun up to 85 million years earlier (Martin et al. 2013). Accordingly, the overlapping dates (given analytical uncertainties) are more a coincidence than a correlation. No two of the three events are established as being "coincident within analytical uncertainty," let alone all three. This is an intolerable abuse of radiometric data.

We have modified this sentence by removing any speculation about a connection between the Rietfontein glaciation and the impact we date. Rather we now state that Yarrabubba 'coincides within uncertainty with the termination of the youngest glacial deposit associated with Palaeoproterozoic snowball Earth conditions, suggesting that the impact site may have been covered by a continental ice sheet'. Also as stated in the response to reviewer 1 above we have removed reference to the Lomagundi carbon isotope excursion from the abstract.

Regarding the final abstract sentence (lines 24-27), the authors have not demonstrated that water vapour release by the impact "provides a viable mechanism for affecting the earth's climate" when the implied climate change is snowball termination. Even the residence time of water vapor in the stratosphere (months to years) is likely too short given the enormous heat sink of a largely glaciated planet. Moreover, ice clouds have albedo as well as quasi-greenhouse effects that are complex and counteracting.

We agree with the reviewer that the interactions of impact produced water vapour with the atmosphere are complex, and a thorough treatment of this aspect is clearly outside the scope of this manuscript which is focused on new geochronology results. Nevertheless, we assert that the results of our impact simulations are valuable for future climate modelling studies of the Palaeoproterozoic Snowball Earth and clearly relevant in the context of the new age we report. Therefore, we have modified the final sentence of the abstract and replaced the statement that impact released water vapour 'provides a viable mechanism for affecting the Earth's climate' which 'provides new input parameters for climate models investigating the mechanisms for termination of Paleoproterozoic Snowball Earth and the role impact cratering may play in modifying Earth's climate.' We feel that this revision of the Abstract to a 'hypothesis to be tested' kind of concluding statement also aligns much better with the more circumspect approach that we have adopted in our revised Discussion, following the reviewer's recommendation from the initially submitted manuscript.

I have not gone over the rest of the revised ms. The abstract is unacceptable to me.

PF Hoffman

We hope the reviewer will find our modifications appropriate, and will consider the rest of the manuscript. We strongly feel this work contains important new scientific data (geochronology and impact modelling)

and at least highlights that the Earth Science community should consider the potential of impacting to affect Earth System Change.

Reviewers' comments:

Reviewer #1 (Remarks to the Author):

The manuscript is much improved by getting rid of references to carbon isotope excursions. However, in trying to address the reviewers concerns that the Riefontien is not necessarily a Snowball Earth event, the authors have created a new problem by evoking a "Paleoproterozoic snowball earth period". It is not a period--there are distinct glaciations and if they are snowballs there is strong hysteresis to a nonglacial state--this is not like Quaternary glacial-interglacial periods. The authors need to just simply state that the age of the impact may overlap with the youngest Paleoproterozoic glacial deposits. Although the age and extent of these glacial deposits is poorly constrained, if the Riefontien deposits reflect snowball Earth conditions, the impact site may have been covered by a continental ice sheet.

On line 23-24 just say: "The age of the Yarrabubba impact event coincides within uncertainty with the the youngest Paleoproterozoic glacial deposits."

Line 142-156: Rewrite. It is not intermittent periods that define Snowball Earth. If it is a Snowball it is a synchronous global event. The 2.2 Ga glacial deposits have not been found elsewhere and it is not clear that it is global. The bit about a lull in geodynamic activity is also nonsense. Long-term, climate is set by the sources and sinks of CO₂ so stating that impacts are more likely than the geological carbon cycle to control longer-term background climate is just weird. It is possible for impact to end an event, like Snowball, but they are not going to change long-term climate state. Again, this is why creating a "Paleoproterozoic snowball Earth period" is not helpful to their presentation. I wish the authors could just state simply and clearly what the record is without repeatedly getting themselves in trouble by being loose and speculative. Just follow Gumsley et al. (2017, PNAS), state the age constraints on the three older Paleoproterozoic glaciations and that there is also a younger deposit in South Africa that may represent a younger Snowball Earth event.

The data and modeling is awesome--don't confuse it with nonsense. Be precise and concise.

Reviewer #2 (Remarks to the Author):

The following statements are incorrect and misleading:

lines 142-3: "Glacial diamictite deposits spanning ca. 2.4 to 2.2 Ga signify intermittent periods of global glacial conditions."

There is paleomagnetic evidence for "global" (i.e., low latitude) glaciation in only a single glacial formation. Scattered glacial deposits do not indicate global glaciation unless that are shown to be of the same age. This has nowhere been demonstrated for Palaeoproterozoic glaciation.

lines 144-5: "the Rietfontien diamictite . . . represents the potential final glacial episode of Paleoproterozoic snowball Earth."

There is no evidence that Rietfontien glaciation was global or that global glaciation occurred at that time.

lines 148-9: "the termination of the Paleoproterozoic snowball period at ca. 2225"

There is abundant and widespread evidence of nonglacial and even tropical conditions between

2225 and 2400 Ma.

lines 163-5: "The deposition of silica ejecta on any pre-existing ice sheets would have resulted in a decrease in albedo, changing the balance of solar radiation and directly initiating melting of ice."

Ejecta from large impacts typically extends out only a few crater-radii in distance. This is a trivial area relative to the planetary surface. Moreover, given the volume of vaporized ice ejected by the impact, any silica dust would be instantaneously covered by snow, limiting its (minimal) effect on planetary albedo to the period when the atmosphere was most opaque, minimizing the effect of surface relative to top-of-the-atmosphere albedo.

Reviewer #1 (Remarks to the Author):

The manuscript is much improved by getting rid of references to carbon isotope excursions. However, in trying to address the reviewers concerns that the Riefontien is not necessarily a Snowball Earth event, the authors have created a new problem by evoking a "Paleoproterozoic snowball earth period". It is not a period--there are distinct glaciations and if they are snowballs there is strong hysteresis to a nonglacial state--this is not like Quaternary glacial-interglacial periods. The authors need to just simply state that the age of the impact may overlap with the youngest Paleoproterozoic glacial deposits. Although the age and extent of these glacial deposits is poorly constrained, if the Reifontein deposits reflect snowball Earth conditions, the impact site may have been covered by a continental ice sheet.

We appreciate the point raised by the reviewer and have modified the text exactly as suggested to avoid any misinterpretation. These changes have been made both within the abstract and within the discussion section on 'Implications for the Palaeoproterozoic Earth'. As suggested by the reviewer, we have removed any reference to a 'snowball earth period' and have explicitly stated that the age and extent of the Rietfontein deposits are poorly constrained (lines 156-157).

On line 23-24 just say: "The age of the Yarrabubba impact event coincides within uncertainty with the the youngest Paleoproterozoic glacial deposits."

We have made the reviewers suggested modification to the abstract. As mentioned above, on line 23-24 we now state: "The age of the Yarrabubba impact coincides, within uncertainty, with the temporal constraint for the youngest Palaeoproterozoic glacial deposits, the Rietfontein diamictite".

Line 142-156: Rewrite. It is not intermittent periods that define Snowball Earth. If it is a Snowball it is a synchronous global event. The 2.2 Ga glacial deposits have not been found elsewhere and it is not clear that it is global. The bit about a lull in geodynamic activity is also nonsense. Long-term, climate is set by the sources and sinks of CO₂ so stating that impacts are more likely than the geological carbon cycle to control longer-term background climate is just weird. It is possible for impact to end an event, like Snowball, but they are not going to change long-term climate state. Again, this is why creating a "Paleoproterozoic snowball Earth period" is not helpful to their presentation. I wish the authors could just state simply and clearly what the record is without repeatedly getting themselves in trouble by being loose and speculative. Just follow Gumsley et al. (2017, PNAS), state the age constraints on the three older Paleoproterozoic glaciations and that there is also a younger deposit in South Africa that may represent a younger Snowball Earth event.

We have thoroughly modified this section on the suggestion of the reviewer, removing the statement of a 'Paleoproterozoic snowball Earth period' and have carefully worded the text. We expressly state that the Rietfontein deposit in South Africa is the youngest glacial deposit in the Palaeoproterozoic, and may represent a snowball Earth event, but that its age is based on the overlying Hekpoort basalt, and that evidence for a Snowball Earth environment is lacking. Moreover, we also state the extent of this glacial deposit is unknown.

As far as our previous citation of a 'geodynamic lull', that concept was published in a 2018 Nature Geoscience paper (Spencer et al.). Rather than focus on 'geodynamic lull', which is ill-defined, in our revision

we now simply cite the specific ~50 Myr window from 2266-2014 Ma that was identified as an apparent hiatus in global magmatism. This is a far more specific and relevant observation, as it directly goes to addressing whether volcanic outgassing is a viable means of ending ice conditions at 2225 Ma. It is unlikely that volcanic outgassing ended ice conditions during a global hiatus in magmatism.

The data and modeling is awesome--don't confuse it with nonsense. Be precise and concise.

We are glad the reviewer appreciates the data and the modelling results and hope these modifications have helped further clarify the manuscript.

Reviewer #2 (Remarks to the Author):

The following statements are incorrect and misleading:

lines 142-3: "Glacial diamictite deposits spanning ca. 2.4 to 2.2 Ga signify intermittent periods of global glacial conditions."

There is paleomagnetic evidence for "global" (i.e., low latitude) glaciation in only a single glacial formation. Scattered glacial deposits do not indicate global glaciation unless that are shown to be of the same age. This has nowhere been demonstrated for Palaeoproterozoic glaciation.

We have modified this sentence 'At least four glacial diamictite deposits, three of which are found on multiple cratons, are recognized between 2.4 to 2.2 Ga³⁸. Of these deposits, the > 2.42 Ga Makganyene diamictite from the Kaapvaal craton of southern Africa has been interpreted to represent low latitude glaciers that may signify a snowball period with global ice conditions³⁹. The youngest Palaeoproterozoic glacial deposit, the Rietfontein diamictite within the Transvaal basin of South Africa has a minimum depositional age of 2225 ± 3 Ma based on the overlying Hekpoort basalt (Fig. 4); which is within analytical uncertainty of the Yarrabubba impact event.'

We also state '...while the extent of the Rietfontein diamictite is unknown, if this deposit reflects snowball Earth conditions....' In an effort to further clarify the situation regarding the youngest glaciation.

lines 144-5: "the Rietfontein diamictite . . . represents the potential final glacial episode of Paleoproterozoic snowball Earth."

There is no evidence that Rietfontein glaciation was global or that global glaciation occurred at that time.

We have removed the statement that the Rietfontein diamictite 'represents the potential final glacial episode of Paleoproterozoic snowball Earth', and stated that the 'The geographic extent of the Rietfontein diamictite is poorly constrained, and it is not yet known if global ice conditions existed at this time.'

lines 148-9: "the termination of the Paleoproterozoic snowball period at ca. 2225"

There is abundant and widespread evidence of nonglacial and even tropical conditions between 2225 and 2400 Ma.

We agree, prior to 2225 Ma climate may have oscillated but included recognized glaciations, which may or may not be global (at least one is reasonably interpreted as global). Our point was that it is widely recognized that only after 2225 Ma is there an extended period of non-glacial conditions. We modified the sentence so it now reads: 'What caused the extended absence of glacial conditions after ca. 2225 Ma is debated.'

lines 163-5: "The deposition of silica ejecta on any pre-existing ice sheets would have resulted in a decrease in albedo, changing the balance of solar radiation and directly initiating melting of ice."

Ejecta from large impacts typically extends out only a few crater-radii in distance. This is a trivial area relative to the planetary surface. Moreover, given the volume of vaporized ice ejected by the impact, any silica dust would be instantaneously covered by snow, limiting its (minimal) effect on planetary albedo to the period when the atmosphere was most opaque, minimizing the effect of surface relative to top-of-the-atmosphere albedo.

While the reviewer is correct that the proximal ejecta blanket only extends ~2.5 crater radii, a large volume of rock will melt and travel much further afield, forming distal ejecta such as spherules. For example the ~100 km diameter Popigai crater located in northern Siberia produced a spherule layer that has been found in the Atlantic and Pacific basins (e.g. Glass and Simonson, 2013). Our point remains that distal impact ejecta would also have some effect on albedo, particular if it falls on ice. However, we have modified this statement to explicitly state that these effects would be short term and localized to the impact environment.

REVIEWERS' COMMENTS:

Reviewer #1 (Remarks to the Author):

The presentation has improved. The dates and modeling are nice contributions. Although I disagree with much of the discussion, at least it is now less speculative and misleading than previous versions.

Reviewer 1:

The presentation has improved. The dates and modeling are nice contributions. Although I disagree with much of the discussion, at least it is now less speculative and misleading than previous versions.

We are glad the reviewer feels that the presentation has improved and that the manuscript is less speculative and misleading. We are pleased they appreciated the quality of the dates and modeling.